# The Combination of Molecular Adjuvant CCL35.2 and DNA Vaccine Significantly Enhances the Immune Protection of *Carassius auratus gibelio* against CyHV-2 Infection

**DOI:** 10.3390/vaccines8040567

**Published:** 2020-10-01

**Authors:** Xingchen Huo, Chengjian Fan, Taoshan Ai, Jianguo Su

**Affiliations:** 1Department of Aquatic Animal Medicine, College of Fisheries, Huazhong Agricultural University, Wuhan 430070, China; Huoxingchen2020@webmail.hzau.edu.cn (X.H.); yc07649@umac.mo (C.F.); 2Laboratory for Marine Biology and Biotechnology, Pilot National Laboratory for Marine Science and Technology, Qingdao 266237, China; 3Engineering Research Center of Green development for Conventional Aquatic Biological Industry in the Yangtze River Economic Belt, Ministry of Education, Wuhan 430070, China; 4Wuhan Chopper Fishery Bio-Tech Co., Ltd., Wuhan Academy of Agricultural Science, Wuhan 430207, China; scs@wuhanagri.com

**Keywords:** *Carassius auratus gibelio*, Molecular adjuvant, DNA vaccine, Immune response, CyHV-2

## Abstract

Cyprinid herpesvirus 2 (CyHV-2) infection results in huge economic losses in gibel carp (*Carassius auratus gibelio*) industry. In this study, we first constructed recombinant plasmids pcORF25 and pcCCL35.2 as DNA vaccine and molecular adjuvant against CyHV-2, respectively, and confirmed that both recombinant plasmids could be effectively expressed in vitro and in vivo. Then, the vaccination and infection experiments (*n* = 50) were set as seven groups. The survival rate (70%) in ORF25/CCL35.2 group was highest. The highest specific antibody levels were found in ORF25/CCL35.2 group in major immune tissues by qRT-PCR, and confirmed in serum by ELISA assay, antibody neutralization titer, and serum incubation-infection experiments. Three crucial innate immune indices, namely C3 content, lysozyme, and total superoxide dismutase (TSOD) activities, were highest in ORF25/CCL35.2 group in serum. pcORF25/pcCCL35.2 can effectively up-regulate mRNA expressions of some important immune genes (IL-1β, IL-2, IFN-γ2, and viperin), and significantly suppress CyHV-2 replication in head kidney and spleen tissues. The minimal tissue lesions can be seen in ORF25/CCL35.2 group in gill, spleen, and trunk kidney tissues by histopathological examination. The results indicated that the combination of DNA vaccine pcORF25 and molecular adjuvant pcCCL35.2 is an effective method against CyHV-2 infection, suggesting a feasible strategy for the control of fish viral diseases.

## 1. Introduction

*Carassius auratus gibelio* (gibel carp) is an important freshwater aquaculture species in China, which has great development potential and market value [1]. However, herpesvirus hematopoietic necrosis disease caused by Cyprinid herpesvirus 2 (CyHV-2) is a fatal contagious disease that results in huge economic losses [2]. CyHV-2 also known as goldfish hematopoietic necrosis virus (GFHNV) or goldfish herpesvirus hematopoietic necrosis virus (HVHNV), is a member of Cyprinivirus, including CyHV-1, CyHV-2, and CyHV-3 [3]. CyHV-2 was firstly reported to cause disease in cultured goldfish in Japan in 1990s, soon afterwards, the disease has been reported around the world [4]. This disease caused massive death of cultured gibel carp with a mortality rate of 90–100% and spread rapidly [3]. The entire genome sequence of CyHV-2 (Accession No. NC_019495) was analyzed and 36 membrane proteins were predicted in 2013 [5]. The envelope glycoprotein ORF25 of CyHV-2 and CyHV-3 demonstrates suitability as components of subunit vaccine [6,7].

Vaccination is the most appropriate way to control infectious diseases. There are three kinds of commonly used fish vaccines, namely live vaccine, inactivated vaccine, and genetic engineering vaccine, which includes DNA vaccine, recombinant subunit vaccine, gene deletion/mutant vaccine, and living-vector vaccine [8]. DNA vaccine enjoys huge advantages, such as the simplicity of delivery, the capacity to express diverse antigens in vivo and no need for cold chain storage [9]. However, many researchers are concerned by the safety issues of DNA vaccine, including the potential to integrate into host cellular DNA and the development of autoimmunity [10]. Safety trials have been conducted and there is no convincing evidence that DNA vaccine can be integrated into host cellular DNA [11,12] and that it has association with autoimmunity development [13,14]. DNA vaccines have been shown to have significant advantages in improving the protection rate against viral infections in previous studies [15], which inspires us to research DNA vaccines.

The intensity of the immune response induced by a DNA vaccine relies on the amount and type of antigen-presenting cells attracted to the sites of vaccination. One feasible strategy for enhancing the immune effect of DNA vaccines is co-delivery of plasmid encoding cell-recruiting chemokine as molecular adjuvant [16]. Molecular adjuvants can recruit a large numbers and types of antigen-presenting cells at the injection site. In this way, the efficiency of antigen presentation can be improved, thereby enhancing the immune response induced by DNA vaccines [17]. Chemokine (C-C motif) ligand (CCL) 4 is a CC chemokine subfamily member defined by the sequential positioning of conserved cysteine residues, and CCL4 was also called macrophage inflammatory protein-1β (MIP-1β) [18]. CCL4 is the most potent chemoattractant of a CD4^+^CD25^+^ T cell population, which is a characteristic phenotype of regulatory T cells, and the recruitment of regulatory T cells to B cells and APCs by CCL4 plays a central role in the normal initiation of T cell and humoral responses [19]. Furthermore, mammalian CCL4 acts as a chemoattractant to monocytes and NK cells through interaction with specific G-protein-coupled receptors (GPCRs) consisting of seven transmembrane domains on these cells [20]. Besides chemotactic activity, receptor binding of mammalian CCL4 also triggers the modulation of downstream effector functions, such as T cell differentiation, dendritic cell maturation, and activation of T cells and granulocytes [21]. Lillard et al. demonstrated that CCL4 can enhance humoral immunity and (CD4) T cell response against ovalbumin in a mouse model for mucosal immunization [20,22]. When used with HER2/neu DNA vaccine, the molecular adjuvant encoding mouse CCl4 can increase the proliferation of T lymphocytes and the production of specific antibody reaction, and inhibit the growth of tumor [20]. All these functions and effects prove the crucial functions of CCL4, so it is of great significance to study CCL4 as a molecular adjuvant. We aligned mammalian CCL4 with 81 chemokines in grass carp (*Ctenopharyngodon idella*) genome [23], and found that *C. idella* CCL35.2 shares highest identity with mammalian CCL4. We retrieved CCL35.2 in *C. gibelio* transcriptomes and employed it as an adjuvant in the present study.

Here, we combined the advantages of molecular adjuvant and DNA vaccine against CyHV-2 infection. We constructed recombinant plasmid pcORF25 and pcCCL35.2 as DNA vaccine and molecular adjuvant respectively. We verified that both molecular adjuvant and DNA vaccine could be expressed in vitro/vivo. After vaccination and challenge, we demonstrated that the CCL35.2/ORF25 group have relatively higher immune efficacy, as well as a better survival rate and tissue protection. CCL35.2 as a molecular adjuvant significantly enhances the effect of DNA vaccine and inactivated vaccine, which also proved the necessity of studying this adjuvant. This study will contribute to a better understanding of the functional mechanism of molecular adjuvant CCL35.2 in combination with DNA vaccine.

## 2. Materials and Methods

### 2.1. Fish, Virus and Cells

Healthy gibel carp, weighting about 35 g, were obtained from a fish farm in Fanzhou, Hubei Province, China. The animals acclimatized for two weeks before experiments, and we fed the gibel carp with commercial feed twice a day. Gibel carp were divided into seven groups with fourteen tanks (*n* = 50), each tank was filled with 300 liters of water, and the water temperature was kept at 24 ± 1 °C. All procedures of animal experiments were approved by the guideline of the Animal Experimental Ethical Inspection of Laboratory Animal Centre, Huazhong Agriculture University. License number of the using of laboratory animal is HZAUFI-2019-016.

CyHV-2 (1 × 10^7^ TCID_50_mL^−1^) and Gibel carp brain (GiCB) cells were kind gifts from professor Linbing Zeng, Yangtze River Fisheries Research Institute, Chinese Academy of Fishery Sciences. We kept the virus at −80 °C. 

The Fathead minnow (FHM) cells were cultured in DMEM (Gibco) supplemented with 10% FBS (Gibco, Beijing, China), 100 U/ml penicillin, and 100 U/ml streptomycin and maintained at 28 °C in a humidified atmosphere of 5% CO_2_. GiCB cells were cultured at 28 °C in M199 medium (Sigma, St. Louis, MO, USA) supplemented with 10% FBS (Gibco), 100 U/ml penicillin, and 100 mg/mL streptomycin.

### 2.2. Preparation of Inactivated Vaccine, DNA Vaccine and Molecular Adjuvant

The inactivated vaccine of CyHV-2 was preserved in our laboratory [24]. Briefly, gibel carp were intraperitoneally injected with 200 μL of virus suspension respectively. When fish showed apparent symptoms, the viscera samples were collected and homogenized with 10 times of PBS (pH = 7.4). The homogenate was frozen at −20 °C and thawed at room temperature after complete freezing. The operation was repeated three times. Then, the homogenate was centrifuged at 12,000 rpm for 30 min. After that, the supernatant was filtered with a 0.45-μm filter membrane. Determination of TCID_50_ was conducted in a 96-well plate. β-propiolactone (BPL) with a final concentration of 0.1% was added to the CyHV-2 suspension, subject to vortexing for 15 min, stood at 4 °C for 72 h, and with hydrolyzing BPL at 37 °C for 4 h. The suspension was used to remove the precipitate produced by BPL at 12,000 rpm. The supernatant was filtered through a 0.22-μm membrane to collect the filtrate. After that, the safety and sterility were examined.

To prepare the DNA vaccine and molecular adjuvant, the complete sequence of CyHV-2-ORF25 and gibel carp CCL35.2 were amplified by PCR using specific primer pairs (Appendix A). *Bam*H I with kozak sequence (GCCACC) and *Eco*R I sites were respectively added to the forward and reverse primers. For the subsequent detection of ORF25 protein, we inserted the HA-tag sequence into the reverse primers. PCR products and pcDNA3.1 (+) were digested with *Bam*H I and *Eco*R I simultaneously. The digestion products were purified and ligated by T4 ligase. pcDNA3.1-ORF25HA (pcORF25) and pcDNA3.1-CCL35.2HA (pcCCL35.2) were confirmed by sequencing, which will be used as DNA vaccine and molecular adjuvant in the following experiments, the same plasmid was also used in in vitro and in vivo transfection experiments. The EndoFree Maxi Plasmid Kit (TIANGEN, Beijing) was used to extract plasmids. The endotoxin content in the preparation was less than 0.5 EU/mL.

### 2.3. Expression of Recombinant pcORF25 and pcCCL35.2 in FHM Cells

FHM cells were cultured in DMEM medium supplemented with 10% fetal bovine serum until reaching 80%–90% confluence. According to the manufacturer’ instructions, transfection agent of Fugene 6 (Promega, Shanghai, China) and plasmids were transfected into FHM cells at a ratio of 2:1 (*v*/*v*), before 4 μg of plasmid DNA was added to the medium in a 6-well plate. After 6 h of incubation, the medium was removed, and the cells were cultured in fresh medium at 28 °C with 5% CO_2_. 36 h after transfection, cells were subsequently fixed with 4% (*w*/*v*) paraformaldehyde for 10 min, permeabilized with 0.1% (*v*/*v*) Triton X-100 for 10 min, and blocked the nonspecific binding with 3% (*v*/*v*) BSA at 37 °C for 1 h. The slides were washed three times with PBS and then incubated with mouse anti-HA antibodies (1:3000, ABclonal) and FITC-conjugated goat anti-mouse IgG (1:200, ABclonal) at 37 °C for 1 h, successively. The nuclei of all cells were stained with 1 mg/mL Hoechst 33,342 at room temperature for 10 min and photographed on UltraVIEW VoX 3D Live Cell Imaging System (PerkinElmer, Waltham, MA, USA). Cells transfected with pcDNA3.1 were used as negative control.

### 2.4. Expression of Recombinant pcORF25 and pcCCL35.2 in Muscle Tissue

The target plasmids were injected into the dorsal muscle of healthy gibel carp (20 μg/animal), respectively. Muscle tissue (0.5–1.0 cm^2^) around the injection site were sampled on the second day. The muscle tissue was made sections, fixed with xylene for 15 min, soaked in ethanol and PBS for 15 min respectively. Next, experimenter heated 0.01 M sodium citrate buffer solution (pH = 6.0) to about 95 °C in water bath, and used the buffer heat the sections for 15 min. The sections were then incubated with 5% BSA at 37 °C for 1 h. After incubation, the sections were bound to primary mouse anti-HA antibodies (1:3000, ABclonal) and secondary antibodies (FITC-conjugated goat anti-mouse IgG, 1:200, ABclonal) at 37 °C for 1 h, respectively, and then stained with 1 mg/ml Hoechst 33,342 (1:1000) for 10 min in a moisture chamber. After washing with PBS three times, the sections were observed on UltraVIEW VoX 3D Live Cell Imaging System (PerkinElmer). Muscle sections from pcDNA3.1 treated fish were used as negative control.

### 2.5. Vaccination, Challenge, Sampling and Survival Assay

Fourteen tanks of gibel carp were divided into seven kinds of experimental groups. There were 100 fishes in each experimental group, and 50 of them were used to measure mortality and the remaining 50 were used for sampling. Seven experimental groups were respectively injected with 10 μg pcORF25/10 μg pcDNA3.1-neo (ORF25 group), 10 μg pcORF25/10 μg pcCCL35.2 (ORF25 + CCL35.2 group), 10 μg pcCCL35.2/100 μL inactivated vaccine (CCL35.2 + Vaccine group), 100 μL inactivated vaccine (Vaccine group), 10 μg pcCCL35.2/10 μg pcDNA3.1-neo (CCL35.2 group), 20 μg pcDNA3.1-neo (pcDNA3.1-neo group), 100 μL PBS (PBS group, pH = 7.4). The recombinant plasmid was solved in 100 μL PBS, and the site of injection was dorsal muscle. CyHV-2 (50 μL, 1 × 10^7^ TCID_50_mL^−1^) challenge was intraperitoneally carried out on 14-day post injection (dpi). The sera from five fish of each group were gathered on 14 dpi before virus challenge for antibody examination. Five fish of each group were sacrificed for harvesting serum, spleen and head kidney tissues on 1, 3, 7, 14, 15, 17, 21, and 28 dpi for serum biochemistry and qRT-PCR, and spleen, trunk kidney, gill tissues were collected from each group on 21 dpi for histopathological examination. The mortality was monitored from 14 to 28 dpi.

### 2.6. Serum Antibody Titer by ELISA Assay

The whole blood from each sample was allowed to clot at 37 °C for 1 h, and then was centrifuged at 1500× *g* for 5 min. The serum samples were collected for the detection of serum antibody titer. The serum antibody titer in grass carp was determined by the enzyme-linked immunosorbent assay (ELISA). Briefly, each well of 96-well plate was coated with CyHV-2. After the plate was incubated overnight at 4 °C, it was washed with PBS containing 0.1% Tween-20 (PBS-T). Then, it was blocked with 1% skimmed milk powder, 0.5% bovine serum albumin and blocking buffer at 37 °C for 2 h. Plate was washed three times with PBST, and fish sera from immunized group and control group were added at dilution of 1:200. The plate was incubated with test sera at 37 °C for 1 h. The plate was washed again and incubated with 1:1000 diluted horseradish peroxidase (HRP) conjugated rabbit anti-fish IgM antibody (Zoonbio Tech Co, Nanjing, China) at 37 °C for 0.5 h. Finally, 0.1 mL 3, 3′, 5, 5′-tetramethylbenzidine (TMB) (Solarbio, Beijing, China) was added in each well and developed for 15 min. The reaction was stopped by adding 50 μL of 2 M sulfuric acid. After the calibration with blank control, the OD value of samples at 450 nm was read with Synergy 4 Hybrid Microplate Reader (BioTek, Winooski, VT, USA).

### 2.7. Serum Antibody Neutralization Tests and Biochemistry Index Assays

Gibel carp were anesthetized with 3-Aminobenzoic acid ethyl ester methanesulfonate (MS222). Blood samples were collected from the caudal vein and were placed for 1 h at room temperature. After centrifugation at 4 °C, 4500 rpm for 15 min, the serum samples were collected and stored at −80 °C. Serum of each group was heat-inactivated at 56 °C for 30 min and diluted at a rate of 1:20 in PBS. The serum neutralization titers were measured by CyHV-2 and GiCB cells according to the previous report [25]. Briefly, a serial 1:2 dilution of the serum was mixed with equal volume of CyHV-2 containing 50% TCID_50_mL^−1^ and incubated at 37 °C for 1 h (the dilution range is between 1:2 to 1:40). We added 100 μL of the diluted serum mixture to each well containing GiCB cells and incubated the plate at 28 °C for 1 h. Then, the mixtures were gently aspirated and 0.2 mL of fresh M199 supplemented with 2% FBS was added back to each well. The 96-well microplates were incubated at 28 °C for 5–7 days. PBS and negative serum were used as negative controls. The disappearance of lesions in each well was regarded as positive indication of neutralization, and neutralization titers were calculated as the reciprocal of the highest serum dilution. 

For the serum-incubation-infection experiment, the diluted serum was gently mixed with 1 × 10^7^ TCID_50_mL^−1^ of virus in a 1:1 vol ratio and then was incubated at 25 °C for 1 h. The 100 μL mixture was injected into gibel carp intraperitoneally. Fish injected with an equal volume of PBS was set as blank control. Dead fish were collected, and spleens were sampled for CyHV-2 detection by RT-PCR. Specific primer pair A040F/A040R (Appendix A) targeting the gene segment of CyHV-2-TK was used for investigation.

The serum biochemical indexes of complement 3 (C3), lysozyme and total superoxide dismutase (TSOD) were assayed by the corresponding commercial kits (Nanjing Jiancheng Bioengineering Institute, Nanjing, China). 

### 2.8. Immune Genes and Virus Gene for qRT-PCR

The total RNAs of head kidney and spleen tissues were isolated with TRIzol reagents (Aidlab, Beijing, China) according to the instruction, the quality of RNA was determined by measuring absorbance at 260 and 280 nm, and its integrity was tested by electrophoresis in 2% agarose gel. mRNAs were reverse-transcribed into cDNAs, respectively, with MMLV reverse transcriptase, RNase inhibitor (Thermo Fisher Scientific, Waltham, MA, USA), and hexamer random primer. The primers for qRT-PCR analyses were listed in Appendix A. Sequence source (GenBank): β-actin (LC382464.1), ORF25 (NC_043042.1), TK (KM200722.1), viperin (AY303809.1), IL-1β (AB757757.1), IFN-γ2 (AB570432.1), and IgM (GU563726.1). In addition, the CCL35.2 and IL-2 of gibel carp were obtained by comparing with grass carp sequences and deposited in GenBank: CCL35.2 (MN338055) and IL-2 (MN338056). β-actin was used as an internal control gene, and the relative mRNA expression was calculated with the 2^−△△CT^ method.

### 2.9. Histopathological Examination

The spleen, trunk kidney and gill tissues were dissected and fixed immediately in 10% neutral buffered formalin for 24 h, dehydrated, paraffin-embedded, and sectioned. Then, 4 μm sectioned samples were mounted on aminopropyl triethoxysilane-coated slides. Following the deparaffinization in xylene, sections were rehydrated, stained with hematoxylin and eosin (HE), and mounted with neutral gum. Then, the images were captured.

### 2.10. Statistical Analysis

The results were expressed as the means ± standard deviation (SD) and all statistical analysis were done using SPSS 26.0 package. The experimental data of each group were subjected to the Kruskal–Wallis test followed by Dunn’s multiple comparison (with Bonferroni adjustment) to identify the significance (*p* < 0.05). The protection rate data were analyzed by Mantel–Cox test.

## 3. Results

### 3.1. Recombinant pcORF25 and pcCCL35.2 Can Express Corresponding Proteins In Vitro and In Vivo

Two pairs of specific primers with kozak consensus sequence (GCCACC) and HA-tag were used to amplifying CCL35.2 and ORF25, respectively. The overexpression vectors were constructed and verified by sequencing (Appendix A). We transfected FHM cells and injected muscle tissue with pcORF25 and pcCCL35.2. CCL35.2HA and ORF25HA protein expressions were confirmed in FHM cells (Figure 1) and gibel carp muscle tissue (Figure 2) by IFA. These results indicated that the CCL35.2HA and ORF25HA proteins successfully express in vitro and in vivo.

### 3.2. pcORF25/pcCCL35.2 Mixture Remarkably Improves the Survival Rate of Gibel Carp

To assess the protective effect of the vaccine and adjuvant, animals in different groups were injected with pcORF25, pcCCL35.2/pcORF25, pcCCL35.2/inactivated vaccine, inactivated vaccine, pcCCL35.2, pcDNA3.1-neo and PBS, respectively. Then they were challenged with CyHV-2 on 14 dpi. Fish survival was monitored and counted in the next 14 days (Figure 3). pcORF25/pcCCL35.2 group demonstrated highest survival rate (70%), which was significantly higher than other groups. Compared with PBS group, DNA vaccine and inactivated vaccine could provide a certain protective effect to gibel carp, but DNA vaccine had higher protective effect. DNA and inactivated vaccines conjugated with adjuvant showed better results. The survival rate of pcDNA3.1-neo group was almost consistent with that of PBS group.

### 3.3. pcORF25/pcCCL35.2 Significantly Enhances Specific Antibody Levels and Neutralization Capacity

In the vaccination and infection experiments, mRNA expressions of antibody IgM significantly increased on 3 dpi (spleen) or 7 dpi (head kidney) in CCL35.2/ORF25 group, and from then on kept highest expression in the seven groups (Figure 4A–C). Then, we examined the antibody IgM protein levels in gibel carp sera (Figure 4D). Both pcORF25 and inactivated vaccine could increase the levels of serum IgM, and the levels of serum IgM were higher when combined with adjuvant. The serum IgM levels in CCL35.2/ORF25 group kept highest since D14, were low in CCL35.2, pcDNA3.1-neo, and PBS groups before challenge, and increased after challenge.

Furthermore, we performed the antibody neutralization tests in sera in vitro and in vivo. The sera were gathered on 14 dpi before virus injection. The serial dilution sera were mixed with CyHV-2 and incubated GiCB cells for neutralizing antibody titer assays. Our results showed that neutralizing antibody titers were significantly high in pcORF25 and inactivated vaccines conjugated with pcCCL35.2 adjuvant, were low in vaccine only, CCL35.2, pcDNA3.1-neo and PBS groups before challenge, and increased after challenge (Figure 5A). In addition, we injected 100 μL of mixture of serum (1:20 dilution) and virus (1 × 10^7^ TCID_50_mL^−1^) equal volume incubation into gibel carp to investigate the protection effect in vivo. The PBS without virus was used as blank control. The PBS group with virus was used as negative control. In the Vaccine, CCL35.2, pcDNA3.1-neo and PBS groups, gibel carp began to die on 3 dpi. In the ORF25 and CCL35.2 + Vaccine groups, fish death occurred on 4 dpi. Meanwhile, death occurrence was delayed on 5 dpi in the CCL35.2/ORF25 group (Figure 5B and Table 1). The protection rates of Vaccine, CCL35.2, and PBS groups were 0% as the same as that of the control group. The protection rates of ORF25, CCL35.2 + Vaccine and Vaccine groups were 20%, 20%, and 10%, respectively. The protection rate of CCL35.2/ORF25 group was the highest among all the experimental groups, reaching 50%. None of the fish in the blank control group died during the experiment and no more fish died after 7 dpi in all groups (Figure 5B). Ten moribund fish were randomly selected for virus detection by specific primers targeting CyHV-2 TK gene. A band of about 324 bp was amplified in each fish (Figure 5C). The fragment was sequenced and CyHV-2 in these fish was confirmed. The enhanced IgM in major immune tissues and serum indicated that pcCCL35.2 with pcORF25 could elicit a robust systemic immune response in gibel carp.

### 3.4. Improvement of Serum Innate Immune Indices

In the detection of serum innate immune indicators, C3 content had a noticeable improvement in CCL35.2/ORF25 mixture group after 3 dpi. The C3 concentration of CCL35.2 + ORF25 group returned to normal at 14 dpi, and increased rapidly after 15 dpi, with a significant difference compared with other groups (Figure 6A). In CCL35.2/ORF25 group, the lysozyme activity rose after vaccination, then decreased slightly, and increased even more after challenge. The lysozyme activity of CCL35.2/ORF25 group increased more rapidly after challenge, and there was significant difference compared with other groups on 17 dpi (Figure 6B). Meanwhile, the TSOD activity increased only in the CCL35.2/ORF25 mixture group and ORF25 group. The TSOD activity in CCL35.2/ORF25 group increased more rapidly (Figure 6C). In general, the combination of DNA vaccine ORF25 and molecular adjuvant CCL35.2 can effectively improve serum innate immune levels.

### 3.5. Immune Gene Expression was Significantly Upregulated by pcORF25/pcCCL35.2 Treatment

mRNA expressions of representative immune genes including IL-1β, IL-2, IFN-γ2 and viperin were examined by qRT-PCR in spleen and head kidney at different time points post vaccination and challenge (Figure 7 and Figure 8). In the CCL35.2, pcDNA3.1-neo and PBS groups, mRNA expressions of the immune genes in head kidney and spleen were not significantly up-regulated. The immune genes in CCL35.2/ORF25, ORF25, CCL35.2/Vaccine, and Vaccine groups were significantly up-regulated, but the up-regulated effect was most obvious in the CCL35.2/ORF25 group. mRNA expressions of IL-1β in CCL35.2/ORF25, ORF25, CCL35.2/Vaccine, and Vaccine groups significantly increased on 1 dpi, and went down after that, and remarkably upregulated on 15 dpi (Day 1 post virus injection) (Figure 7A,B). With the change of IL-1β expression, mRNA expressions of IL-2, IFN-γ2 and viperin subsequently rose and fallen obviously (Figure 7C,D and Figure 8A–D). These results showed that innate immunity and adaptive immunity were evidently enhanced by pcCCL35.2/pcORF25 treatment.

### 3.6. pcORF25/pcCCL35.2 Treatment Effectively Relieves the Tissue Lesions against Virus Infection

To assess the extent of tissue lesion in each group, gill, spleen and trunk kidney tissues were dissected, fixed and sliced for HE staining on 21 dpi (Figure 9). Vaccine, CCL35.2, pcDNA3.1-neo and PBS groups showed the most severe lesions including vacuolization, karyorrhexis, hypertrophied nuclei and sloughing of the epithelial cells. By contrast with the healthy tissue, CCL35.2/ORF25 group had evidently protective effect, ORF25 and CCL35.2/Vaccine group had some protective effect. These results suggested that pcCCL35.2/pcORF25 treatment can effectively protect fish tissues from lesions. Detailed statistical results of tissue lesions were presented in Appendix A.

### 3.7. Virus Replication Was Suppressed in CCL35.2/ORF25 Group

Three treatments have significantly inhibitory effect on the replication of the virus in head kidney, including ORF25, CCL35.2/ORF25, and CCL35.2/Vaccine groups. CCL35.2/ORF25 group was the most effective in inhibiting CyHV-2 (Figure 10A). In spleen, only CCL35.2/ORF25 group showed significant suppression at all the time points (Figure 10B). The results indicated that molecular adjuvant/DNA vaccine can indeed inhibit the proliferation of CyHV-2.

## 4. Discussion

Considering economic, environmental, and ethical factors, as well as protective effect, vaccination against disease is currently the most appropriate way for aquaculture to control pathogens [26]. DNA vaccine, as the third generation vaccine, has clear advantages in the prevention and control of immunization in major diseases in aquaculture [27]. It is essential to screen suitable protective antigen epitopes for the development of DNA vaccines. Herein, 36 ORFs encoding for the membrane proteins of CyHV-2 were analyzed by a bioinformatics technique, among which the ORF25 protein had the best immunogenicity, and a better protection rate was obtained when using DNA vaccine encoding ORF25 [5]. In another experiment, 20 μg truncated proteins of ORF25, ORF25C and ORF25D were used to immunize healthy gibel carp, after challenge, protection rates were 75%, 63%, 54% respectively [6]. Molecular adjuvant plasmids expressing cytokines, chemokines or costimulatory molecules can be used together with antigenic DNA vaccine plasmids. The cells transfected with the molecular adjuvant plasmid secreted adjuvant to the surrounding area, stimulating local antigen presenting cells (APC) and cells draining lymph nodes. [9]. The combined immunization with rabies virus glycoprotein DNA vaccine and CCL4 improved the immune response of mice, confirming the potential of CCL4 as a DNA vaccine adjuvant [28]. Previous study has shown that the constructed dsDNA vaccine with VAA as antigen gene and CCL4 as immune adjuvant has potential application value in the protection of Japanese flounder eel plague [17]. Therefore, complete ORF25 membrane protein of CyHV-2 was used as antigen to prepare DNA vaccine, and chemokine CCL35.2 (mammalian CCL4 analogue in gibel carp) was used as molecular adjuvant to enhance the immune effect of DNA vaccine in the present study.

The pcDNA3.1 plasmid was selected as a vector for DNA vaccine and molecular adjuvant. Previous studies have demonstrated that CMV promoter could drive the expression of exogenous genes in transfected fish cell lines and immunized fish muscle [29,30]. In the present study, IFA assays showed that FHM cell lines and gibel carp muscle transfected/injected pcORF25 and pcCCL35.2 plasmids, and expressed corresponding proteins. 

The protection rate is the best reflection of the vaccine efficacy. In our previous study, the immersion immunization with an inactivated vaccine had a 26% protection rate and a 14% increase after the addition of β-glucan [24]. Previous other report showed that the relative survival rates of goldfish immunized with formalin-inactivated CyHV-2 were 42.5% and 57.6%, challenged at 4 and 8 weeks post vaccination, respectively [31]. In the present study, ORF25 group demonstrated a 52% protection rate, while the relative survival rate was just 38% in the β-propiolactone inactivated vaccine group. When the molecular adjuvant CCL35.2 was added to the DNA vaccine group and the inactivated vaccine group, the protection rates were increased by 18% and 12%, respectively. The survival rate strongly suggested that our developed DNA vaccine pcORF25 conjugated with molecular adjuvant CCL35.2 could be a rather attractive strategy to improve the viability of fish. Moreover, all the treatment groups displayed immune protection to some extent, although the various survival rates could be affected by the experimental conditions, methods, etc.

Antibody plays a central role in the humoral immune responses. Among the three subtypes of Immunoglobulin (Ig), IgM is widely present in the systemic circulation [32]. In our study, mRNA expression of IgM was upregulated after vaccine injection and rapidly upregulated after challenge. The rapid upregulation of IgM mRNA suggested that memory B cells can be activated fleetly [33]. The enhanced serum IgM levels and titers indicated that pcCCL35.2/pcORF25 could elicit a robust humoral immune response in vaccinated fish. The result indicated that molecular adjuvant CCL35.2 combined with DNA vaccine ORF25 could remarkably promote humoral immunity to achieve the better protection effect of fish. Most effective vaccines induce high levels of specific antibodies that is crucial in determining the quality of subsequent adaptive immune responses [34]. In our results, pcCCL35.2/pcORF25 treatment could induce strong immune responses and high titers of neutralizing antibodies post vaccination and challenge. The fish mortality was evidently relieved in the group injected virus incubated with pcCCL35.2/pcORF25 group serum, indicating that CCL35.2/ORF25 group serum had better neutralizing efficacy to CyHV-2.

Serum innate immune indices often reflect the immune levels in animals. The previous study suggested that resistance against bacterial and viral pathogens in grouper correlates with increases in complement content, SOD, and lysozyme activities [35]. The complement system is one of the main components of non-specific humoral immunity, which plays a major role in labeling and removing exogenous microorganisms, as well as in regulating the inflammatory response and specific immune response [36]. After vaccination and challenge, the content of complement 3 in serum increased rapidly. These results suggested that C3 is actively involved in antigen labeling, enhancing specific and nonspecific immune responses to eliminate antigens. In addition, superoxide dismutase and lysozyme are closely related to the immune levels in fish [37]. After the vaccination and challenge, TSOD and lysozyme showed an upward trend. The results indicated that CCL35.2/ORF25 induces salutary immune responses in blood.

It is necessary to investigate the mechanisms of vaccine on innate immunity and adaptive immunity. IL-1 is an important early inflammatory cytokine that plays a vital role in regulating defensive and pathological innate immunity responses [38]. IL-1β was rapidly upregulated after vaccine injection and challenge in this study. After the treatment with CCL35.2/ORF25, mRNA expression of IL-1β was rapidly up-regulated, indicating that this treatment method can better improve the early inflammatory response. IL-2 promotes peripheral blood leukocytes (PBL) proliferation, sustains high level expression of CD4^+^ and CD8^+^, and has a stronger effect on the upregulation of the T helper 1 (Th1) cytokine (IFNγ1, IFNγ2, TNFα2, and IL-12) [39]. Moreover, IL-2 produced by T cells promotes the proliferation and survival of activated T cells and is necessary to activate natural killer (NK) cells and B cells to synthesize immunoglobulin [40,41]. In the present study, IL-2 was most significantly upregulated with CCL35.2/ORF25 injection in spleen and head kidney, suggesting CCL35.2/ORF25 promoted Th1 pathway in cellular immunity. Interferons (IFNs) are secretory cytokines with important and complex immune functions. In the pcCCL35.2/pcORF25 treatment group, IFN-γ2 was rapidly upregulated on the third day after challenge. Viperin is an antiviral effector that can effectively inhibit DNA and RNA virus replication [42,43]. Previous studies have found that viperin expression can be up-regulated by direct stimulation of virus, bacteria or stimulation of IFN-like factor [44]. Viperin expression in the CCL35.2/ORF25 group remained at a high level, indicating that this method could enable gibel carp to establish a barrier against virus and effectively fight against CyHV-2 infection.

Furthermore, we examined the protective effects of vaccines on tissue lesions after challenge. In previous study, histopathological sections could intuitively show the pathological injury of CyHV-2 infection, and gill, spleen and kidney will have a large pathological reaction in the course of CyHV-2 infection [45]. CyHV-2 generates herpes-like virus nucleocapsid in the nucleus and generates multiple encapsulated viral particles aggregates in the cytoplasm, which damage tissue functions through replication of virus [46]. In a histological assay, CCL35.2/ORF25 group showed slightest tissue lesions, which also indicated that pcCCL35.2/pcORF25 treatment effectively protected tissue function from destruction. 

Finally, mRNA expression of CyHV-2 (TK gene) was investigated to verify whether virus replication was inhibited by the vaccines. Compared with other groups, CyHV-2 replication in ORF25/CCL35.2 group in head, kidney, and spleen tissues was inhibited to a greater extent.

## 5. Conclusions

DNA vaccine (CyHV-2 ORF25) conjugated with molecular adjuvant CCL35.2 (mammalian CCL4 analogue) remarkably improved the survival rate of gibel carp against CyHV-2 infection (a fulminating infectious disease). This strategy can effectively promote the specific antibody levels, titers, and virus neutralization capacity in important immune tissues and serum, serum innate immune indices, and mRNA expressions of immune regulation and effector genes, relieve the main target tissue lesions, and suppress the virus replication. The present study develops an efficient strategy for a novel vaccine to conquer viral diseases.

## Figures and Tables

**Figure 1 vaccines-08-00567-f001:**
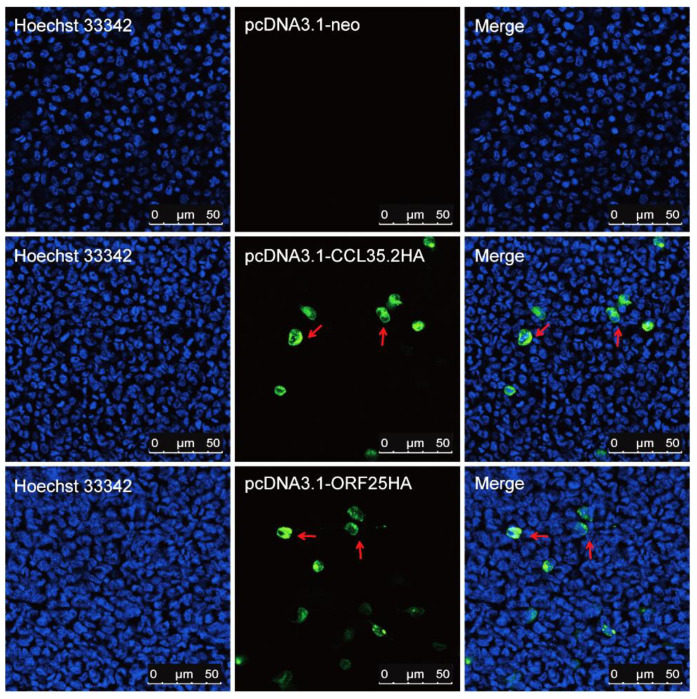
Detection of CCL35.2HA and ORF25HA protein in vitro. FHM cells were transfected with pcDNA3.1-neo, pcCCL35.2 and pcORF25, respectively. IFAs were performed with mouse anti-HA antibody and goat anti-mouse IgG-FITC (green) second antibody. Cell nuclei were stained by DAPI (blue). The red arrow indicates the specific green fluorescence produced by indirect immunoassay.

**Figure 2 vaccines-08-00567-f002:**
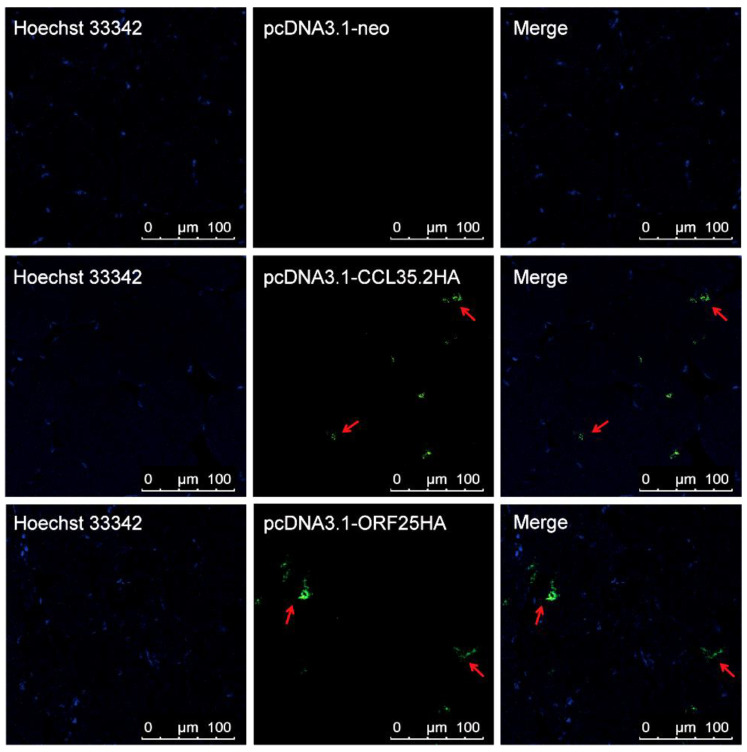
Detection of CCL35.2HA and ORF25HA protein in vivo. The gibel carp were intramuscularly injected with pcDNA3.1-neo, pcCCL35.2 and pcORF25, respectively. The muscle sections were carried out IFAs with mouse anti-HA antibody and goat anti-mouse IgG-FITC (green) second antibody. Cell nuclei were stained by DAPI (blue). The red arrow indicates the specific green fluorescence produced by indirect immunoassay.

**Figure 3 vaccines-08-00567-f003:**
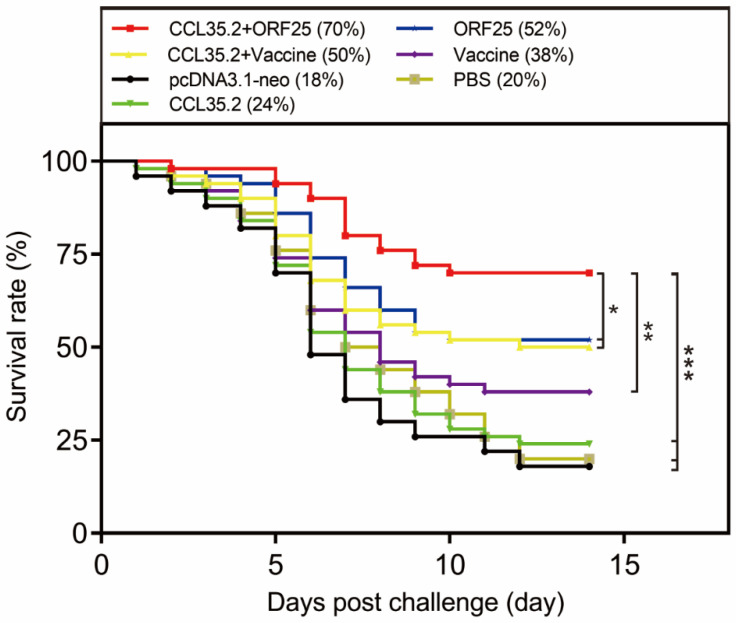
Survival rates of gibel carp against CyHV-2 infection. Animals were intramuscularly injected with pcORF25, pcCCL35.2/pcORF25, pcCCL35.2/Vaccine, Vaccine, pcCCL35.2, pcDNA3.1-neo, PBS on day 0. Fish in each group (*n* = 50) were challenged with 50 μL CyHV-2 (1 × 10^7^ TCID_50_mL^−1^) on day 14, and death events in each group were monitored on the next 14 days. *p* values were calculated by Log–rank (Mantel–Cox) Test (* *p* < 0.05, ** *p* < 0.01, *** *p* < 0.001).

**Figure 4 vaccines-08-00567-f004:**
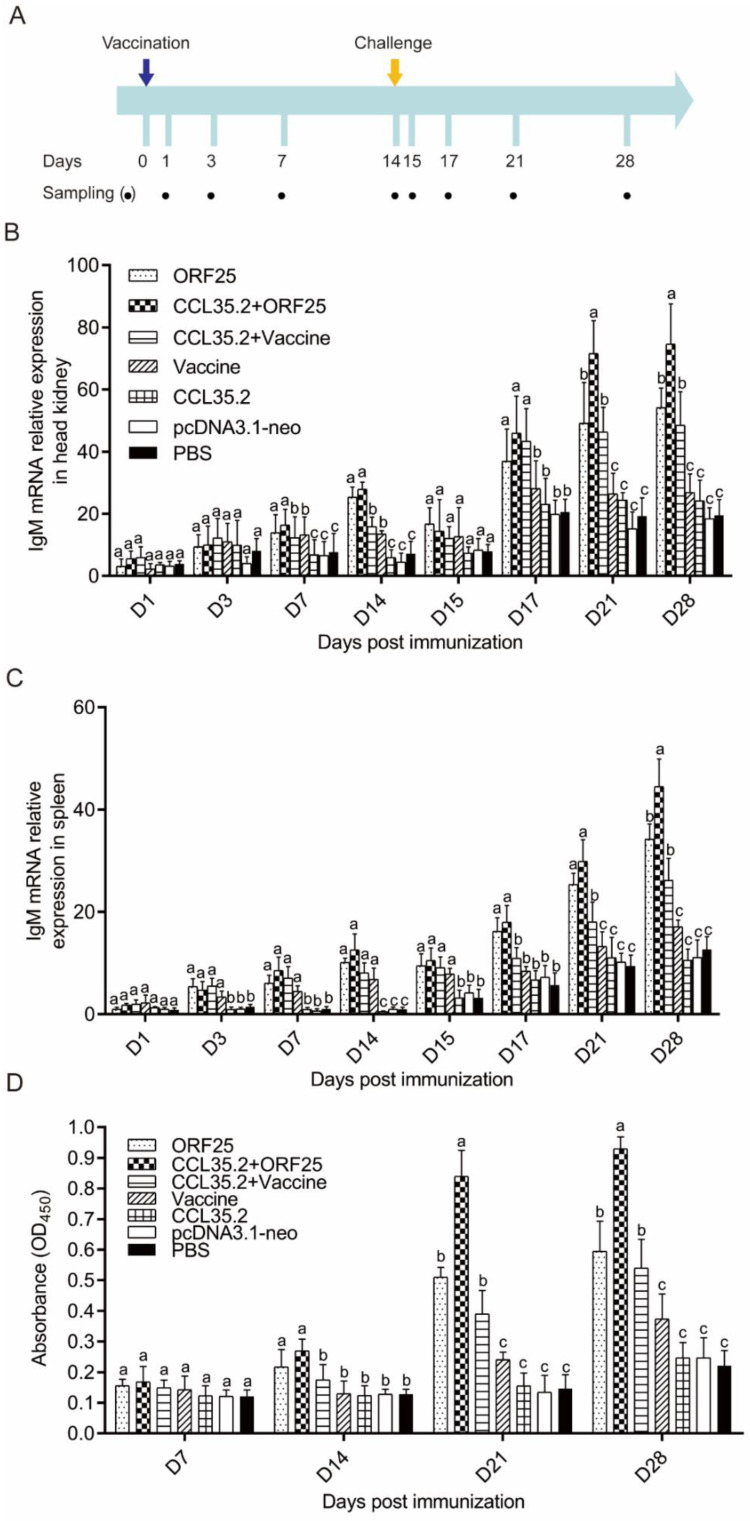
IgM levels in tissue and serum. (**A**) Vaccination/challenge and sampling schedule. mRNA expression patterns of IgM in head kidney (**B**) and spleen (**C**) were determined by qRT-PCR. β-actin gene was used as an internal control gene. (**D**) Detection of serum antibody levels by ELISA in gibel carp. Different superscript letters in each group (a–c) denote significant variations suggested by the Kruskal–Wallis statistics at 95% of significance, followed by the Dunn test with Bonferroni adjustment as the post hoc test (*p* < 0.05).

**Figure 5 vaccines-08-00567-f005:**
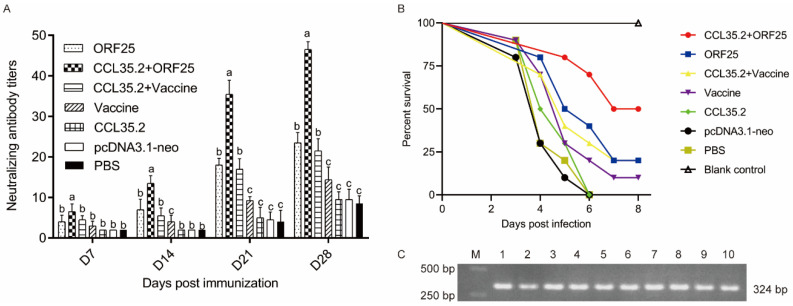
The neutralization assay of anti-CyHV-2 serum in vitro and in vivo. (**A**) The reciprocals of the highest serum dilutions prevent cytopathic effect in each experimental group and at different time-points. Different superscript letters in each group (a–c) denote significant variations suggested by the Kruskal–Wallis statistics at 95% of significance, followed by the Dunn test with Bonferroni adjustment as the post hoc test (*p* < 0.05). (**B**) Each group diluted serum (1:20, 50 μL) was gently mixed with 1 × 10^7^ TCID_50_mL^−1^ of virus in a 1:1 vol ratio. The 100 μL mixture was injected into gibel carp intraperitoneally. Fish injected with an equal volume of PBS was set as blank control. Death was monitored daily until no more fish died. (**C**) RT-PCR detection of CyHV-2. 1–10: random ten moribund fish; M: marker.

**Figure 6 vaccines-08-00567-f006:**
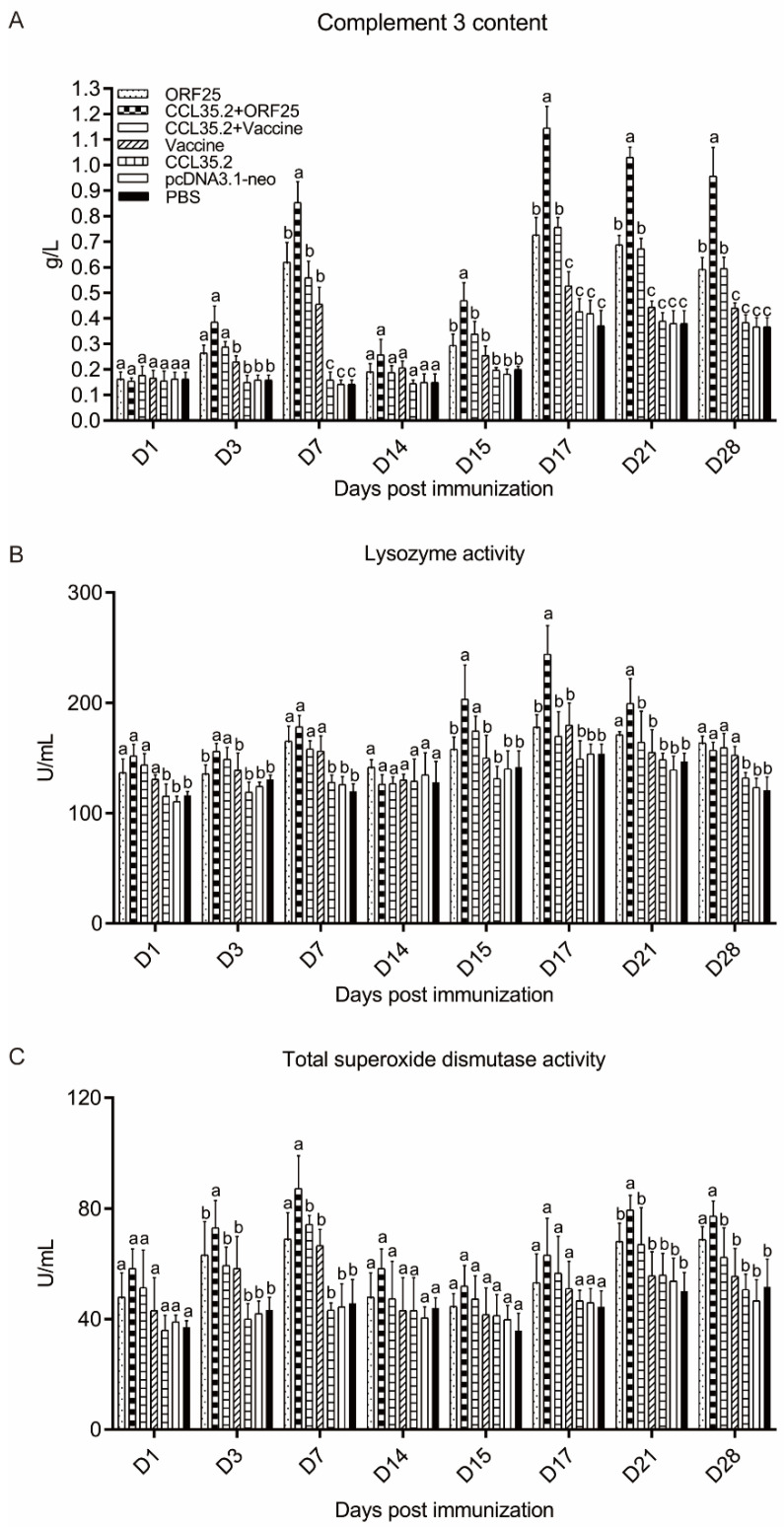
Serum innate immune indices. Complement 3 (**A**), lysozyme (**B**) and total superoxide dismutase (**C**) were determined by the commercial kits (Nanjing Jiancheng Bioengineering Institute, Nanjing, China). Different superscript letters in each group (a–c) denote significant variations suggested by the Kruskal–Wallis statistics at 95% of significance, followed by the Dunn test with Bonferroni adjustment as the post hoc test (*p* < 0.05).

**Figure 7 vaccines-08-00567-f007:**
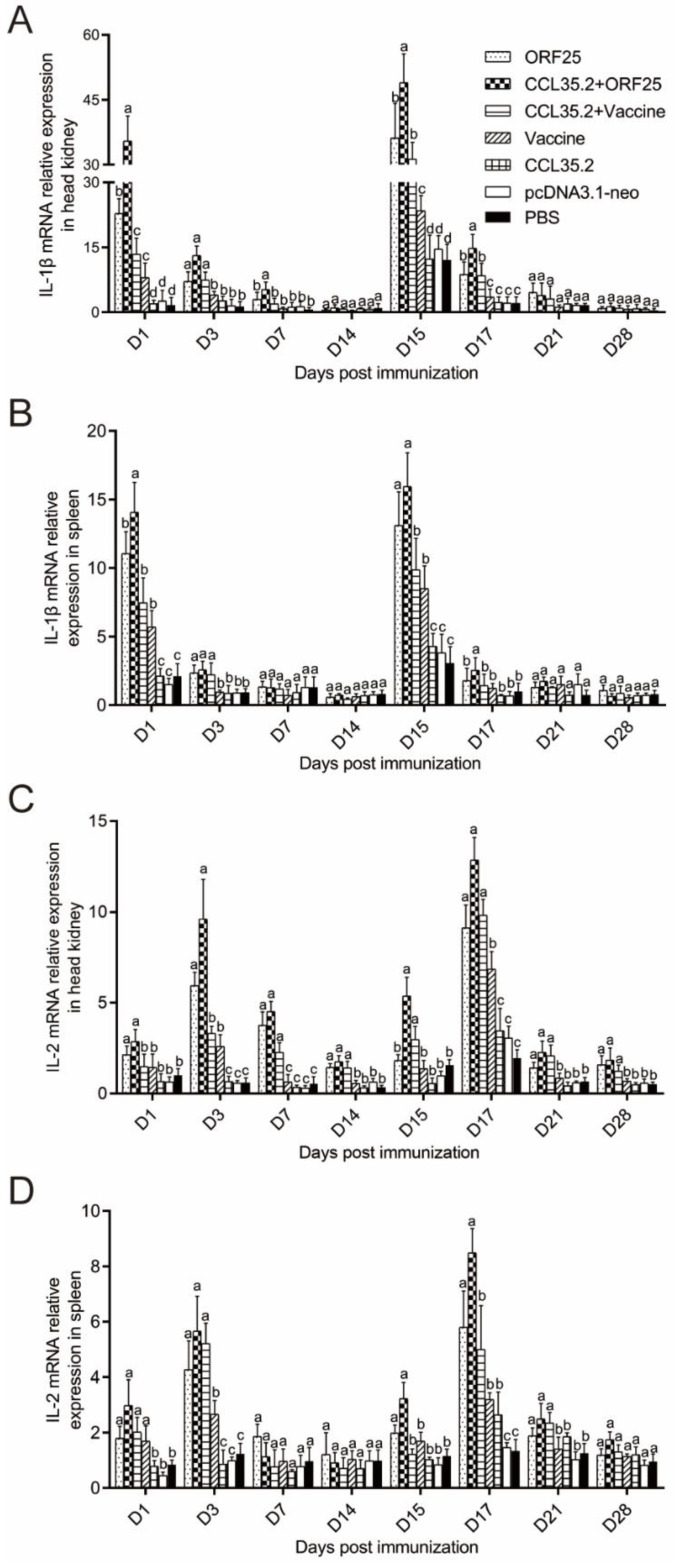
mRNA expression patterns of representative important immune regulation and effector genes in head kidney and spleen tissues. IL-1β (**A**,**B**), IL-2 (**C**,**D**) transcripts were examined by qRT-PCR. β-actin gene was used as an internal control gene. Different superscript letters in each group (a–d) denote significant variations suggested by the Kruskal–Wallis statistics at 95% of significance, followed by the Dunn test with Bonferroni adjustment as the post hoc test (*p* < 0.05).

**Figure 8 vaccines-08-00567-f008:**
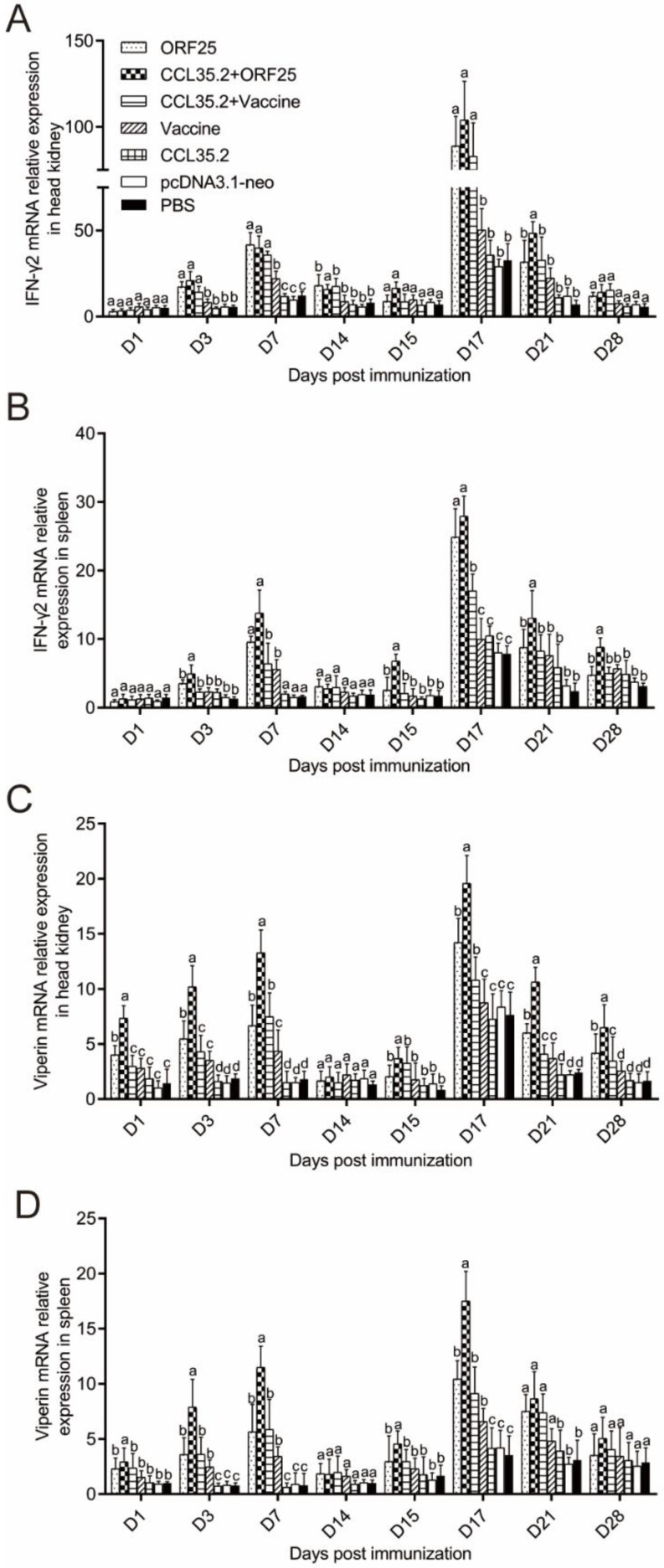
mRNA expression patterns of representative important immune regulation and effector genes in head kidney and spleen tissues. IFN-γ2 (**A**,**B**) and viperin (**C**,**D**) transcripts were examined by qRT-PCR. β-actin gene was used as an internal control gene. Different superscript letters in each group (a–d) denote significant variations suggested by the Kruskal–Wallis statistics at 95% of significance, followed by the Dunn test with Bonferroni adjustment as the post hoc test (*p* < 0.05).

**Figure 9 vaccines-08-00567-f009:**
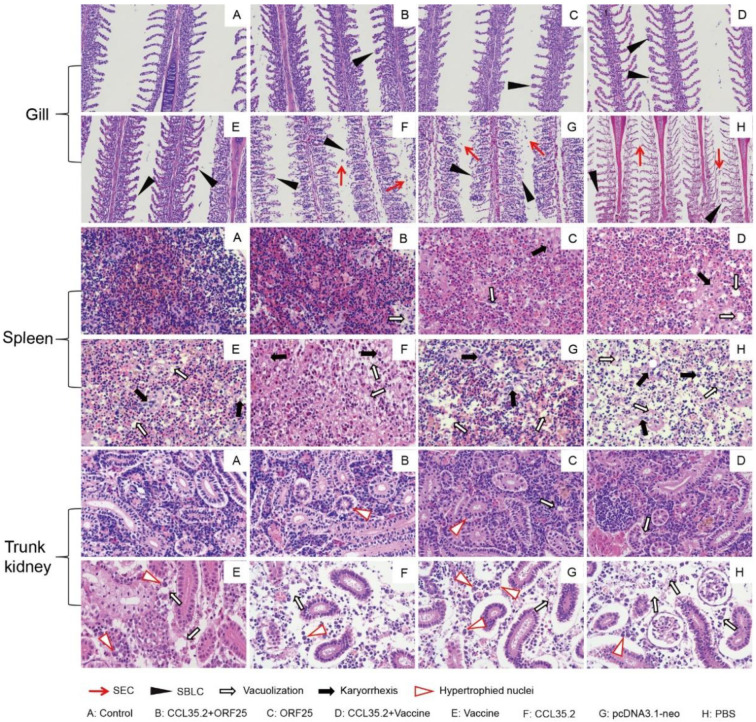
Tissue lesions of gill, spleen and trunk kidney post vaccination and challenge. Gibel carp were intramuscularly injected with pcCCL35.2/pcORF25, pcORF25, pcCCL35.2/Vaccine, Vaccine, pcCCL35.2, pcDNA3.1-neo, and PBS on Day 0, and intraperitoneally injected with CyHV-2 on 14 dpi. Gill, spleen and trunk kidney tissues were collected on 21 dpi, made slides, then stained by HE. Healthy gibel carp tissues were used as blank control. In gill, sloughing of the epithelial cells (SEC) and secondary branchial lamella curl (SBLC) were two main signs. In spleen, vacuolization and Karyorrhexis were two main symbols. In trunk kidney, hypertrophied nuclei and vacuolization were two main features.

**Figure 10 vaccines-08-00567-f010:**
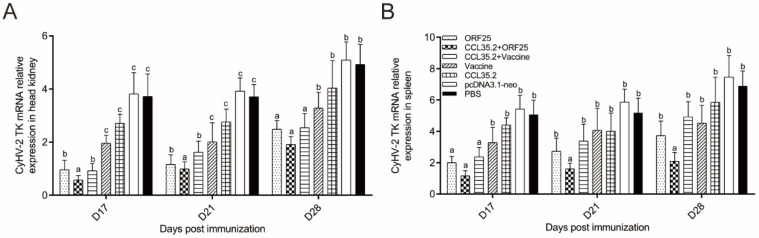
mRNA expression profiles of TK in head kidney (**A**) and spleen (**B**). They were determined by qRT-PCR. β-actin gene was used as a reference gene. Different superscript letters in each group (a–c) denote significant variations suggested by the Kruskal–Wallis statistics at 95% of significance, followed by the Dunn test with Bonferroni adjustment as the post hoc test (*p* < 0.05).

**Table 1 vaccines-08-00567-t001:** Virus neutralization by anti-CyHV-2 antiserum in gibel carp.

Group	Dilution	Number of Survivors					
		2	3	4	5	6	7
CCL35.2/ORF25	1:20	10	10	10	8	5	5
ORF25	1:20	10	10	8	5	4	2
CCL35.2/Vaccine	1:20	10	10	7	6	3	2
Vaccine	1:20	10	9	7	3	2	1
CCL35.2	1:20	10	9	5	3	0	0
pcDNA3.1-neo	1:20	10	8	3	1	0	0
PBS	1:20	10	9	3	2	0	0
Blank control	1:20	10	10	10	10	10	10

Five animals were randomly selected and gathered serum on 14 dpi before virus challenge. 50 μL virus (1 × 10^7^ TCID_50_mL^−1^) was mixed and incubated with 50 μL serum (1:20) and injected into healthy fish. In blank control, each fish was injected with 100 μL PBS without virus.

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
