# Peer review of "The Combination of Molecular Adjuvant CCL35.2 and DNA Vaccine Significantly Enhances the Immune Protection of *Carassius auratus gibelio* against CyHV-2 Infection"

_vaccines, 2020, doi:10.3390/vaccines8040567_

Round 1

Reviewer 1 Report

In this manuscript, Huo et al discuss a novel adjuvent and DNA vaccine to protect against CyHV-2 infection in gibel carp. Whilst this study offers some promising data, several aspects should be addressed before publication.

  • CCL4 should be defined.
  • As CCl4 has been used as a vaccine adjuvent before, these papers should be referenced and discussed more in the intro/discussion.
  • The IF images in Fig 1 and 2 should all be shown at the same magnification so that they can be directly compared. There is no reason given by the authors why they are at different magnifications (the scales are different) and this should not be the case.
  • In Fig.2 - is this how the DAPI staining normal looks in this tissue? It also looks like a DAPI staining from dead/dying cells.
  • In Fig 4 - if this is ELISA data, the pg/mL should be shown instead of the OD.
  • In Fig 4 - the graphs should be made bigger.
  • The analytes assessed in Fig 6 should be described in section 3.4 before presentation of the results.
  • In Fig 7 - the graphs should be made bigger.

Author Response

Dear reviewer,

Thanks for your review and comments on our manuscript. The mistakes of the words and grammars were corrected as your suggestions. Our responses to your concerns are recorded below.

Point 1: CCL4 should be defined.

Response 1: Thank you for your professional comments and constructive suggestion. Following your careful suggestion, we have added a clearer definition of CCL4 to the introduction. Details are as follows: Chemokine (C-C motif) ligand (CCL) 4 is a CC chemokine subfamily member defined by the sequential positioning of conserved cysteine residues, and CCL4 was also called macrophage inflammatory protein-1β (MIP-1β) [18] (Line 63-66).

Point 2: As CCL4 has been used as a vaccine adjuvent before, these papers should be referenced and discussed more in the intro/discussion.

Response 2: Thank you for your professional suggestion. As a vaccine adjuvant, CCL4 has been used in mouse animal models and other fish animal models. More references and discussions were added in the Introduction and Discussion section. Details are as follows: “Lillard et al. demonstrated that CCL4 can enhance humoral immunity and (CD4) T cell response against ovalbumin in a mouse model for mucosal immunization [22]. When used with HER2 / neu DNA vaccine, the molecular adjuvant encoding mouse CCl4 can increase the proliferation of T lymphocytes and the production of specific antibody reaction, and inhibit the growth of tumor [20].” was added in Introduction section (Line 73-77). “Previous study has shown that the constructed dsDNA vaccine with VAA as antigen gene and CCL4 as immune adjuvant has potential application value in the protection of Japanese flounder eel plague [17].” was added in Discussion section (Line 397-399).

Point 3: The IF images in Fig 1 and 2 should all be shown at the same magnification so that they can be directly compared. There is no reason given by the authors why they are at different magnifications (the scales are different) and this should not be the case.

Response 3: Thank you for your valuable advice. We have respectively chosen the same proportion of images to display in Fig 1 and 2 according to your valuable suggestions. In Fig 1, we have replaced the photographs s of indirect immunoassay on CCL35.2HA protein. In Fig 2, we selected the same proportion of control photographs for comparison.

Point 4: In Fig.2 - is this how the DAPI staining normal looks in this tissue? It also looks like a DAPI staining from dead/dying cells.

Response 4: Thank you for your professional suggestion. The nuclear stain we used was Hoechst 33342, a stain with a similar effect to DAPI. In muscle tissue, the dyeing effect of Hoechst 33342 shown in Fig 2 is normal. The staining effect of Xu et al. on muscle nuclei of fish was consistent with ours.

Xu, H.; Xing, J.; Tang, X.; Sheng, X.; Zhan, W. Immune response and protective effect against Vibrio anguillarum induced by DNA vaccine encoding Hsp33 protein. Microb Pathog 2019, 137, 103729, doi:10.1016/j.micpath.2019.103729.

Point 5: In Fig 4 - if this is ELISA data, the pg/mL should be shown instead of the OD.

Response 5: Thank you for your professional comments and constructive suggestion. The results of ELISA for IgM are shown in Fig 4D. However, commercial ELISA kits and protein standards are not available, so we cannot replace OD with pg/mL by drawing standard curve. In the study of Zhu et al, OD value of samples can also reflect the change of IgM.

Zhu, B.; Zhang, C.; Zhao, Z.; Wang, G. Targeted delivery of mannosylated nanoparticles improve prophylactic efficacy of immersion vaccine against fish viral disease. Vaccines 2020, 8, doi:10.3390/vaccines8010087.

Point 6: In Fig 4 - the graphs should be made bigger.

Response 6: Thank you for your constructive suggestion. We changed the arrangement of the graphs in Fig 4 for a clearer display. The schedule of the experiment and the detection data for IgM are shown more clearly.

Point 7: The analytes assessed in Fig 6 should be described in section 3.4 before presentation of the results.

Response 7: Thank you for your professional suggestion. A more detailed analysis of the detection results of C3 content, lysozyme activity and TSOD activity were added in Results 3.4. Details are as follows: “The C3 concentration of CCL35.2+ORF25 group returned to normal at 14 dpi, and increased rapidly after 15 dpi, with a significant difference compared with other groups”, “The lysozyme activity of CCL35.2/ORF25 group increased more rapidly after challenge, and there was significant difference compared with other groups on 17 dpi” and “The TSOD activity in CCL35.2/ORF25 group increased more rapidly” were added in Results section 3.4 (Line 315-322).

Point 8: In Fig 7 - the graphs should be made bigger.

Response 8: Thank you for your constructive suggestion. Figure.7 was divided into Figure.7 including IL-1β (A, B), IL-2 (C, D) and Figure.8 including IFN-γ2 (A, B), viperin (C, D) according to your professional suggestion. The details of qRT-PCR date in Figure7 and Figure 8 are shown more clearly.

Reviewer 2 Report

Huo et al. developed DNA vaccine systems using ORF25 antigen combined with CCL35.2 expressing plasmids for gibel carp against CyHV-2 infection. However, many concerns, especially the experimental design including statistical analyses, need to be clarified before publication.

Major comments:

  1. Please describe how the inactivated vaccine of CyHV-2 prepared in the Materials and Methods section.
  2. Please describe how the plasmids for in vitro and in vivo experiments used in this study were prepared. Also, please state the endotoxin content in the preparations in the Materials and Methods section.
  3. Throughout the manuscript, the authors used only plasmids for transfection experiments. For me, without any polymer (meaning only plasmids), the transfection efficiency is too low, especially in vitro experiments. Please add some explanation.
  4.  Figure 2: Expression duration after transfections is critical. Did you check this?
  5. Figure 3: Please include statistical analysis in the figure.
  6. In all figures, the t-test is not suitable for these kinds of data. Please re-analyze using the Mantel-Cox test for survival and the Kruskal-Wallis test for the others.
  7. How did you determine the plasmid dose (20 ug)?

Minor comments:

  1. Line 223: "in vitro" and "in vivo" should be italic fonts.

Author Response

Dear reviewer,

Thanks for your review and comments on our manuscript. The mistakes of the words and grammars were corrected as your suggestions. Our responses to your concerns are recorded below.

Point 1: Please describe how the inactivated vaccine of CyHV-2 prepared in the Materials and Methods section.

Response 1: Thank you for your professional comments and constructive suggestion. More detailed methods for preparing inactivated vaccines are added in Materials and Methods 2.2. Details are as follows: The inactivated vaccine of CyHV-2 was preserved in our laboratory [24]. Briefly, gibel carp were intraperitoneally injected with 200 μL of virus suspension respectively. When fish showed apparent symptoms, the viscera samples were collected and homogenized with 10 times of PBS (pH=7.4). The homogenate was frozen at - 20 oC and thawed at room temperature after complete freezing. The operation was repeated three times. Then centrifuged at 12000 rpm for 30 min. After that, the supernatant was filtered with 0.45 μm filter membrane. Determination of TCID50 in 96-well plate. β-propiolactone (BPL) with a final concentration of 0.1% was added to the CyHV-2 suspension, vortexing 15 min, standing in 4 oC for 72 h, and hydrolyzing BPL in 37 oC for 4 h. The suspension was used to remove the precipitate produced by BPL at 12000 rpm. The supernatant was filtered through a 0.22 μm membrane to collect the filtrate. After that, the safety and sterility were examined (Line 108-117).

Point 2: Please describe how the plasmids for in vitro and in vivo experiments used in this study were prepared. Also, please state the endotoxin content in the preparations in the Materials and Methods section.

Response 2: Thank you for your professional comments. In our study, plasmids used for in vitro and in vivo experiments were the same, and more detailed plasmids preparation methods were added to Materials 2.2. The kit (EndoFree Maxi Plasmid Kit) we used to extract plasmids can remove endotoxin, and the endotoxin content in preparations can be reduced below 0.5 EU/mL by using endotoxin remover in the kit. The concentration of endotoxin in the plasmid preparation is added to Materials and Methods 2.2. Details are as follows: To prepare the DNA vaccine and molecular adjuvant, the complete sequence of CyHV-2-ORF25 and gibel carp CCL35.2 were amplified by PCR using specific primer pairs (Table S1). BamH I with kozak sequence (GCCACC) and EcoR I sites were respectively added to the forward and reverse primers. For the subsequent detection of ORF25 protein, we inserted the HA-tag sequence into the reverse primers. PCR products and pcDNA3.1 (+) were digested with BamH I and EcoR I simultaneously. The digestion products were purified and ligated by T4 ligase. pcDNA3.1-ORF25HA (pcORF25) and pcDNA3.1-CCL35.2HA (pcCCL35.2) were confirmed by sequencing, which will be used as DNA vaccine and molecular adjuvant in the following experiments, the same plasmid was also used in in vitro and in vivo transfection experiments. The EndoFree Maxi Plasmid Kit (TIANGEN, Beijing) was used to extract plasmids. The endotoxin content in the preparation was less than 0.5 EU/ml (Line 118-128).

Point 3: Throughout the manuscript, the authors used only plasmids for transfection experiments. For me, without any polymer (meaning only plasmids), the transfection efficiency is too low, especially in vitro experiments. Please add some explanation.

Response 3: Thank you for your valuable suggestion. In the in vitro experiments, Fugene 6 was used as transfection reagent, and this detail was added in the Materials and methods section. The distribution of nucleus and fluorescence could not be clearly seen in the position with more fluorescence in the field of view. In order to better display the fluorescence effect, we chose a clearer position in the field of view for the shooting. In the in vivo experiment, we did not use the transfection reagent, because the cost would be considered in the practical application of fish DNA vaccine, and the addition of in vivo transfection reagent would increase the cost of DNA vaccine. In order to provide more accurate data for adjuvants and vaccines in practical applications, we chose not to add transfection agents.

Point 4: Figure 2: Expression duration after transfections is critical. Did you check this?

Response 4: Thank you for your professional comments and constructive suggestion. We took the immunized muscle tissues at 24h, 48h and 72h for indirect immunofluorescence detection. Through IFA assays, we found that the fluorescence intensity was higher in the muscle tissue after 48 h of immunization, so we chose to detect the muscle tissue after 48 h of immunization in the formal experiment.

Point 5: Figure 3: Please include statistical analysis in the figure.

Response 5: Thank you for your constructive suggestion. Through mantel-Cox test, we analyzed the significant differences between CCL35.2/ORF25 and other groups, then presented the important analysis results in Figure 3.

Point 6: In all figures, the t-test is not suitable for these kinds of data. Please re-analyze using the Mantel-Cox test for survival and the Kruskal-Wallis test for the others.

Response 6: Thank you for your constructive suggestion. We used Mantel-Cox test to re-analyze the protection rate data and show the significant difference in Fig. 3. We re-analyzed other data through Kruskal-Wallis test in IBM SPSS software, and followed by the Dunn test with Bonferroni adjustment as the post hoc test. The new data analysis has been reorganized into figures (Fig 3, 4, 5, 6, 7, 8 and 10). And the analysis method of experimental data was added to Materials and Methods 2.8.

Point 7: How did you determine the plasmid dose (20 μg)?

Response 7: According to Liu et al., injection of 10 μg co-expressed plasmid (pIRES-ORF25-IL-1β) per carp was more effective in enhancing immunity than injection of 2 μg co-expressed plasmid, but the injection of 40 μg per tail did not significantly improve immunity. Therefore, we selected 10 μg DNA vaccine and 10 μg molecular adjuvant per tail for immunization.

Liu, L.; Gao, S.; Luan, W.; Zhou, J.; Wang, H. Generation and functional evaluation of a DNA vaccine co-expressing Cyprinid herpesvirus-3 envelope protein and carp interleukin-1 beta. Fish Shellfish Immunol 2018, 80, 223-231, doi:10.1016/j.fsi.2018.05.046.

Round 2

Reviewer 2 Report

I have gone through the revised manuscript, and I think all my comments have been adequately addressed. Thank you again for the opportunity to review this manuscript.